# Androgen and Estrogen β Receptor Expression Enhances Efficacy of Antihormonal Treatments in Triple-Negative Breast Cancer Cell Lines

**DOI:** 10.3390/ijms25031471

**Published:** 2024-01-25

**Authors:** Belen Crespo, Juan Carlos Illera, Gema Silvan, Paula Lopez-Plaza, María Herrera de la Muela, Miriam de la Puente Yagüe, Cristina Diaz del Arco, Maria Jose Illera, Sara Caceres

**Affiliations:** 1Department Animal Physiology, Veterinary Medicine School, Complutense University of Madrid (UCM), 28040 Madrid, Spain; belencre@ucm.es (B.C.); gsilvang@vet.ucm.es (G.S.); pauulopez4@gmail.com (P.L.-P.); mjillera@vet.ucm.es (M.J.I.); sacacere@ucm.es (S.C.); 2Obstetrics and Gynecology Department, Hospital Clinico San Carlos, Instituto de Salud de la Mujer, Instituto de Investigación Sanitaria del Hospital Clínico San Carlos (IsISSC), 28040 Madrid, Spain; maria.herrera@salud.madrid.org; 3Department of Public and Maternal Child Health University, School of Medicine, Complutense University of Madrid, 28040 Madrid, Spain; mdelapuenteyague@yahoo.es; 4Department of Surgical Pathology, Hospital Clínico San Carlos, 28040 Madrid, Spain; crisdelarco@gmail.com

**Keywords:** androgen receptor, estrogen receptor beta, steroid pathway, triple-negative breast cancer

## Abstract

The triple-negative breast cancer (TNBC) subtype is characterized by the lack of expression of ERα (estrogen receptor α), PR (progesterone receptor) and no overexpression of HER-2. However, TNBC can express the androgen receptor (AR) or estrogen receptor β (ERβ). Also, TNBC secretes steroid hormones and is influenced by hormonal fluctuations, so the steroid inhibition could exert a beneficial effect in TNBC treatment. The aim of this study was to evaluate the effect of dutasteride, anastrozole and ASP9521 in in vitro processes using human TNBC cell lines. For this, immunofluorescence, sensitivity, proliferation and wound healing assays were performed, and hormone concentrations were studied. Results revealed that all TNBC cell lines expressed AR and ERβ; the ones that expressed them most intensely were more sensitive to antihormonal treatments. All treatments reduced cell viability, highlighting MDA-MB-453 and SUM-159. Indeed, a decrease in androgen levels was observed in these cell lines, which could relate to a reduction in cell viability. In addition, MCF-7 and SUM-159 increased cell migration under treatments, increasing estrogen levels, which could favor cell migration. Thus, antihormonal treatments could be beneficial for TNBC therapies. This study clarifies the importance of steroid hormones in AR and ERβ-positive cell lines of TNBC.

## 1. Introduction

Breast cancer is one of the most common malignancies, being the main cause of death in women [1]. Breast cancer is classified into different subtypes depending on the expression of the estrogen receptor alpha (ERα), progesterone receptor (PR) and overexpression of the human epidermal growth factor receptor 2 (HER-2) [2]. Thus, there are three main different subtypes of breast cancer: hormone-receptor-positive subtype, the HER-2-positive subtype and the triple-negative breast cancer (TNBC) subtype [3], and therapeutic regimens may differ among subtypes [4]. Endocrine therapies for receptor-positive breast cancer are very common [5]. Those that express ERα and PR can be treated with estrogen receptor modulators (SERMs) or aromatase inhibitors [6]. However, the breast cancer that overexpresses HER-2 can be treated with trastuzumab and pertuzumab, both humanized monoclonal antibodies against HER-2 [7]. In contrast, TNBC subtype is characterized by the lack of expression of ERα and PR and no overexpression of HER-2 [8,9]. Therefore, endocrine therapies have not been contemplated for TNBC treatment [10], being a challenge because of the lack of efficient therapeutic targets [11].

TNBC is a heterogenic breast cancer subtype characterized by different molecular profiles [12]. The TNBC classification by Lehmann and colleagues (2016) is well known [13]. This classification is based on genetic profiles dividing TNBC into basal-like 1 (BL1), basal- like 2 (BL2), immunomodulatory (IM), mesenchymal (M), mesenchymal stem like (MSL) and luminal androgen receptor (LAR) subtypes [13,14]. Although TNBC is characterized by the lack of expression of ERα, PR and HER-2 overexpression, this breast cancer subtype is capable of expressing other receptors like the androgen receptor (AR) or the estrogen receptor β (ERβ) [8,15,16]. The AR is present in around 10–43% of cases of TNBC [8,15,17]. Specifically, the LAR subtype is characterized by a high AR expression [13] and, therefore, this subtype may benefit from antihormonal therapies [18]. In addition, several studies have postulated that AR expression could favor tumor progression in TNBC [19]. Thus, the AR has been postulated to be a promising target for future therapies against TNBC [11]. On the other hand, ERβ has been of interest because of its presence in around 30% of TNBC cases [15,16]. The ERβ role in TNBC is controversial among research; however, it has been observed to be clearly involved in TNBC development processes [20].

The main ligands must be present to exert their action by binding to their receptors within the cells [8]. Androgens and estrogens are the ligands of the AR and ERβ, respectively, and have been found in high concentrations in breast tumors [8]. They are sex steroids derived from cholesterol and they are mainly secreted from the ovaries and the adrenal cortex [21]. Cholesterol can be converted to dihydroepiandrostenedione (DHEA) and androstenedione (A4), which are the main steroid precursors of androgens and estrogens [16]. The 3β-hydroxysteroid dehydrogenase (3βHSD) enzyme converts DHEA in A4 and the 17β-hydroxysteroid dehydrogenase (17βHSD) converts T from A4 [16]. The aromatase enzyme is involved in estrogen synthesis, converting A4 or testosterone (T) to estrone (E1) or 17β-estradiol (E2), respectively, E2 being the main ligand of ER, including ERα and ERβ isoforms [16,21]. On the other hand, the 5α-reductases (5αR) are a family of enzymes involved in the conversion of T to dihydrotestosterone (DHT), which is the main ligand of AR [22].

Although endocrine therapies have not been contemplated for TNBC treatment, in previous studies, it has been observed that this tumor subtype is capable of secreting steroid hormones and, together with the presence of AR and ERβ, could be an indicator that endocrine therapies may influence on tumor development [23,24]. Thus, the inhibition of steroid hormone synthesis by inhibiting the aromatase and 5αR enzymes, could prevent the activation of AR or ERβ [25], exerting a beneficial effect in TNBC treatment. Moreover, there are several studies involving AR antagonist therapies that have shown promising results in TNBC [10,26]. However, fewer studies exist on steroid hormone inhibition and their impact on TNBC development [23,24].

Dutasteride is a 5αR inhibitor that blocks the T to DHT conversion [27] and is widely used for benign prostatic hyperplasia [28]. This drug has a slower metabolism compared with other drugs, remaining a longer time in blood and producing a greater intracellular response [27]. On the other hand, the American Society of Clinical Oncology (ASCO) recommends as initial adjuvant therapy the anti-aromatase drugs after another therapy with tamoxifen in the initial stages of breast cancer [29]. Anastrozole is a nonsteroidal aromatase inhibitor that is widely used in postmenopausal women with ER-positive breast cancer, achieving a good prognosis [30]. In addition, the 17βHSD5 could be overexpressed in some breast cancers and this has been related with breast cancer relapse [31]. ASP9521 is the first selective inhibitor of 17βHSD5 and has been studied for castration-resistant prostate cancer (CRPC) [32]. In addition, previous studies in canine inflammatory triple-negative breast cancer have shown that the use of ASP9521 in Balb/SCID mice is able to reduce tumor progression, so ASP9521 could be beneficial for the treatment of TNBC [24].

Thus, despite TNBC not expressing ERα, PR and HER-2 overexpression, it is capable of secreting steroid hormones and being influenced by hormonal fluctuations [23,24]. Therefore, the aim of this study was to evaluate the effect of antihormonal treatments, dutasteride, anastrozole and ASP9521, on cell viability, proliferation and migration in in vitro processes in various human TNBC cell lines.

## 2. Results

### 2.1. Immunofluorescence Assay

One hundred cells in each field from three different fields were evaluated. Both receptors were expressed in 100% of the cells evaluated in all cell lines. However, the difference in receptor expression intensity between cell lines is clearly noticeable. MDA-MB-453, MDA-MB-468 and MCF-7 expressed AR with greater intensity than the other cell lines studied. In MDA-MB-453, the AR intensity was localized either in the nucleus or cytoplasm, whereas AR expression in MDA-MB-468 and MCF-7 appeared to be higher in the cytoplasm. In addition, SUM-159 also expressed the AR markedly, especially in the nucleus. Finally, SUM-149 and MDA-MB-231 obtained a slight AR expression and it was mainly a cytoplasmic expression (Figure 1A).

On the other hand, all cell lines expressed ERβ (Figure 1B). The expression was mainly cytoplasmic, highlighting in the cytoplasmic membrane. However, the nucleus was lightly stained in all cell lines, being almost null in MCF-7 cells. This receptor expression was more intense in MCF-7, MDA-MB-468 and SUM-159 at the cytoplasmic membrane. In contrast, SUM-149, MDA-MB-231 and MDA-MB-453 expressed ERβ at low intensity both in the cytoplasm and nucleus.

The fluorescence quantification results (Figure 1C) clarified that cells with higher expression of AR—MDA-MB-453, MDA-MB-468 and MCF-7—obtained significant differences (*p* < 0.001) with respect to those with lower AR expression, SUM-149, SUM-159 and MDA-MB-231. On the other hand, significant differences (*p* < 0.001) were also obtained between MDA-MB-468, SUM-159 and MCF-7, which obtained higher ERβ expression, and SUM-149, MDA-MB-231 and MDA-MB-453, which obtained lower ERβ expression.

### 2.2. Sensitivity Assay

To perform this assay, the concentrations of dutasteride, anastrozole and ASP9521 used ranged from 1 *×* 10^−^^5^ M to 6.4 *×* 10^−^^10^ M. The results obtained indicate the dose of drugs to which the cells were sensitive and are summarized in Figure 2. The assay has showed similar half-maximal effective concentration (EC_50_) for each cell line.

In general, almost all cell lines were more sensitive to dutasteride and anastrozole than to ASP9521. In contrast, MCF-7 was more sensitive to ASP9521 than to the other drugs. After 72 h of assay, all cell lines showed similar effective range of treatment, 1 × 10^−^^10^ M. It can be observed that SUM-149 obtained slightly lower EC_50_ values, so this cell line could be more sensitive to antihormonal treatments. On the other hand, MDA-MB-453 and MDA-MB-468 exhibited higher resistance to dutasteride and ASP9521. From the results obtained, the R square of each compound was considered and standardized to the same EC_50_ for all cell lines and drugs, using an EC_50_ of 1 µM.

### 2.3. Cell Viability Assay

All treatments, dutasteride, anastrozole and ASP9521, promote a reduction in SUM-149, SUM-159, MDA-MB-231 and MDA-MB-453 cell viability (Figure 3). However, only SUM-159 and MDA-MB-453 showed significant reductions (*p* < 0.05) with all drugs. The highest decrease can be observed in MDA-MB-453 dutasteride-treated cells with a cell viability reduction of 37.89%. The reductions with the other treatments and in the other cell lines were similar, around 20%. In contrast, anastrozole and ASP9521 treatments resulted in a proliferation increase in MDA-MB-468 and MCF-7 cell lines but without significant differences (*p* > 0.05).

### 2.4. Wound Healing Assay

Treatments produced different effects among cell lines in the wound healing assay. The only cell lines that reduced significantly (*p* < 0.05) their cell migration using dutasteride and anastrozole were SUM-149 and MDA-MB-468 (Figure 4A,E). In contrast, MDA-MB-453 (Figure 4D,F) and MCF-7 increase cell migration slightly with all treatments. SUM-159 increases cell migration with dutasteride and anastrozole, this increase being significant (*p* < 0.05) with dutasteride, although, with ASP9521, a decrease in cell migration was found (Figure 4B). Finally, MDA-MB-231 (Figure 4C) experienced an increase in cell migration with ASP9521 and slightly decreased with dutasteride but no significant differences (*p* > 0.05) were achieved.

### 2.5. Hormonal Levels in Culture Media

Each cell line responded differently to the treatments applied. All hormone levels in culture media are summarized in Figure 5. Focusing on each drug effect, we found that dutasteride treatment generally produced slight increases in T but decreases in DHT levels. In addition, this was accompanied by increases in estrogen levels (E1 and E2) and decreases in A4 and DHEA precursors in some cell lines. Specifically, results showed that T levels in MDA-MB-468 and MCF-7 were significantly increased (*p* < 0.001). In addition, DHT levels were reduced in SUM-159 (*p* < 0.001), MDA-MB-453 (*p* = 0.002) and MCF-7 (*p* < 0.001). Regarding estrogen levels, dutasteride produced a significant decrease in E1 levels in SUM-149 (*p* < 0.001), SUM-159 (*p* = 0.03) and MDA-MB-468 (*p* < 0.001) and a significant increase in MDA-MB-231 (*p* = 0.046), MDA-MB-453 (*p* < 0.001) and MCF-7 (*p* < 0.001) with respect to the control group. In addition, E2 levels were significantly decreased in MDA-MB-231 (*p* < 0.001) and MCF-7 (*p* = 0.021). Also, A4 levels were significantly increased in SUM-159 (*p* < 0.001) and significantly decreased in SUM-149 (*p* < 0.001) and MDA-MB-231 (*p* < 0.001). Finally, DHEA levels were significantly decreased in SUM-159 (*p* = 0.003), MDA-MB-231 (*p* = 0.001) and MDA-MB-453 (*p* = 0.013), and only in MCF-7 (*p* < 0.001) were DHEA levels significantly increased (*p* < 0.001).

Regarding anastrozole treatment, a generalized reduction in estrogen and DHT levels was found among cell lines, while T levels increased in some cell lines. In addition, the use of anastrozole produced different changes between precursors in each cell line. In general, when A4 raised, DHEA dropped or remained at similar levels to the control and, conversely, when DHEA levels increased, A4 levels decreased or were similar to the control. Particularly, the significant differences obtained were as follows. The only cell line that showed significant reductions for both estrogens (E1 (*p* < 0.001) and E2 (*p* < 0.001) was SUM-149. MDA-MB-468 (*p* = 0.021) and MDA-MB-231 (*p* < 0.001) only decreased E2 levels, while, in SUM-159 (*p* = 0.029) and MDA-MB-453 (*p* = 0.010), an increased in E1 levels was found. Regarding androgen levels, significant increases in T levels were found in SUM-149 (*p* = 0.004), MDA-MB-231 (*p* = 0.003) and MCF-7 (*p* < 0.001). In addition, significantly decreased DHT levels were observed in SUM-149 (*p* < 0.001), SUM-159 (*p* = 0.003), MDA-MB-453 (*p* < 0.001) and MCF-7 (*p* < 0.001) cell lines. Results from A4 levels revealed significant increases in SUM-159 (*p* < 0.001) and MDA-MB-453 (*p* < 0.001), contrary to MDA-MB-231 (*p* < 0.001) and MDA-MB-468 (*p* < 0.001), where a decrease in A4 levels was found. Finally, DHEA levels were increased in MDA-MB-468 (*p* < 0.001) and MCF-7 (*p* = 0.004) and decreased in MDA-MB-231 (*p* = 0.007) and MDA-MB-453 (*p* < 0.001).

Similar to anastrozole results, ASP9521 treatment produced a reduction in estrogen levels and a reduction in DHT levels. However, precursor levels differed between cell lines, finding both increases and decreases in A4 and DHEA. Specifically, there were significant reductions in estrogen levels (E1 and E2) in SUM-149 (*p* < 0.001 and *p* = 0.012), SUM-159 (*p* = 0012 and *p* = 0.029) and MDA-MB-453 (*p* = 0.012 and *p* = 0.007). In MDA-MB-231, only E2 levels were significantly reduced (*p* < 0.001) and, in MCF-7, only E1 levels were significantly increased (*p* < 0.001). Results from androgens revealed that T levels were increased in MCF-7. SUM-159 showed reductions in T and DHT levels (*p* < 0.001 and *p* = 0.003), and, in SUM-149 (*p* = 0.014), MDA-MB-453 (*p* < 0.001) and MCF-7 (*p* < 0.001), DHT levels were decreased. Indeed, SUM-159 (*p* = 0.005), MDA-MB-453 (*p* = 0.001) and MCF-7 (*p* < 0.001) increased significantly A4 levels, and MDA-MB-231 (*p* < 0.001) and MDA-MB-468 (*p* = 0.001) decreased significantly. On the other hand, DHEA increase their levels in SUM-149 (*p* = 0.005), MDA-MB-468 (*p* = 0.001) and MCF-7 (*p* = 0.016) and decreased their levels in MDA-MB-453 (*p* = 0.010).

## 3. Discussion

TNBC remains a clinically important challenge [33]. Various biomarkers have been studied as targets for TNBC treatment but systemic chemotherapy and radiotherapy are the only valid therapeutic options to date [34]. Due to the lack of expression of ERα, PR and HER-2 overexpression, antihormonal therapies are not considered in this tumor subtype [10,35]. However, previous studies have stated that neoplastic TNBC cell lines, despite not expressing ERα and PR, are able to produce and secrete steroid hormones [23]. These hormones may exert their action by binding to other receptors such as the AR, ERβ or GPER-1, among others [8,36]. GPER-1 is present in half of TNBCs and its presence has been associated with a metastasis increase and worse prognosis [37,38]. Also, it has been studied that AR is expressed in 32% of TNBC cases [39] and it may be involved in tumor development [21]. On the other hand, although the alpha subunit of the ER is not expressed in TNBC, the beta subunit can be expressed in this type of tumor and could influence, like the AR, in the development of the disease [40]. However, the role of ERβ and AR in TNBC has not been fully clarified and remains controversial among researchers [11,40].

AR has been related to promoting tumor proliferation when ERα is not expressed [16,18], as it has been studied that AR influences in PI3K/AKT pathways [41]. Also, AR expression has been associated with increased metastasis in TNBC by activation of ZEB-1 or the SRC/PI3K pathways [42]. Thus, AR could be an interesting therapeutic target in the treatment of TNBC [43]. On the other hand, ERβ has been associated with both proliferative and antiproliferative effect [44,45], depending on several factors such as mutations in *TP53* [46] or even on the ERβ isoform that is predominantly expressed [20]. In addition, some studies have observed that co-expression of both receptors regulates tumor progression differently than when only one of these receptors is expressed [16,47]. Song et al. (2017) stated that TNBC-AR+ cell lines that did not express ERβ produced more metastasis than those that co-expressed ERβ and AR [48]. Therefore, studying the co-expression of both receptors can be useful to understand the role of them in TNBC.

Therefore, in this study, AR and ERβ expression was studied in TNBC and MCF-7 cell lines. Results revealed that all cell lines co-expressed AR and ERβ but differed in the intensity of expression. Results revealed that MDA-MB-468 and MCF-7 were the two cell lines that expressed both AR and ERβ with great intensity. MDA-MB-453, MDA-MB-468 and MCF-7 expressed AR with higher intensity than the rest of the cell lines, while ERβ was expressed in MDA-MB-468 and MCF-7 cell lines with high intensity. This is in line with numerous authors who have classified MDA-MB-453 as a luminal androgen receptor cell line (TNBC-LAR) [18]. However, SUM149 and MDA-MB-231 slightly expressed both receptors. AR was mostly expressed in the nucleus of MDA-MB-453, whereas, in MDA-MB-468 or MCF-7, it was mostly expressed in the cytoplasm; also, ERβ was mostly expressed in the cytoplasm and cytoplasmic membrane, according to Shaaban et al. (2008), who stated that, in breast cancer, ERβ expression in the cytoplasm is more abundant than in the nucleus [49]. Another study linked the cytoplasmic expression of ERβ to high-grade breast cancer tumors, favoring tumor aggressiveness through non-genomic pathways and due to the influence of estrogens [50]. ERβ is modulated by E2, so the hormonal environment, together with the expression localization of this receptor in the cell, may be a critical factor for the development of the disease [49]. Estrogens, in particular E2, and androgens, especially DHT, are the main ligands of ERβ and AR, respectively [22]. The binding of the ligand to AR and ERβ produces the translocation of the receptor from the cytoplasmic membrane to the nucleus, exerting genomic and non-genomic activities [15]. Taken together, it can be suggested that cell lines that expressed both receptors, such as MDA-MB-453, would be more sensitive to antihormonal treatments than those cell lines that slightly expressed them, such as SUM149. Therefore, the study of AR and ERβ expression and the steroid hormone production could be of great importance to elucidate the behavior of TNBC cell lines [23,51].

In this study, different drugs were used to inhibit steroid hormone production. Dutasteride is a 5αR inhibitor widely used for benign prostatic hyperplasia [26]. On the other hand, anastrozole is an aromatase inhibitor that is used as a first-line adjuvant endocrine therapy for ER+ breast cancer [52]. Finally, ASP9521 is the first 17βHSD-specific inhibitor to be used in human prostatic cancer trials [53]. Our results claim that, the same as hormonal-receptor-positive breast cancer being affected by hormonal imbalance [54], TNBC cell lines may also be affected by this imbalance despite the absence of ERα and PR expression and due to the expression of AR and ERβ.

TNBC cell lines were more sensitive to dutasteride and anastrozole treatments that inhibit the formation of DHT and E2, respectively. The androgen and estrogen inhibition could alter the steroid pathway, modifying AR and ERβ activation and producing changes in cell proliferation and migration. Moreover, it can be observed that cell lines such as MDA-MB-453 and SUM-159 that expressed AR and ERβ at high intensity reduced proliferation when antihormonal treatments were applied compared to those cell lines that slightly expressed these receptors. This was the case of MDA-MB-231, which expressed a low intensity of AR and ERβ and was more resistant to antihormonal treatments in terms of proliferation and migration. Interestingly, although all cell lines obtained similar sensitivities, differences in cell proliferation and migration were observed between cell lines. In line with the AR and ERβ results, SUM-159 and MDA-MB-453 were the most affected cell lines by antihormonal treatments, denoting a reduction of around 20% in cell proliferation. However, the greatest cell migration reduction was observed in SUM-149. Nevertheless, in contrast to TNBC cell lines, the ER-positive cell line MCF-7 was not affected by the studied hormonal treatments. Some authors claimed that some breast cancer cell lines, such as MCF-7, could develop resistance to antihormonal treatments that compromise the drug effectiveness [55].

MDA-MB-453 showed cell viability reduction with dutasteride in around 40% approximately. As it is a TNBC-LAR cell line and expresses AR in abundance [18], this cell line could be closely influenced by androgens. On the other hand, SUM-159, which expressed AR too, reduced their cell viability in 17.5% with dutasteride. It can be observed that DHT levels in MDA-MB-453 and SUM-159 cells treated with dutasteride decreased. This DHT decrease may affect AR activation and, therefore, promote a cell proliferation reduction, as some authors state that anti-androgenic treatments in TNBC-AR+ subtypes were beneficial for patients [10]. Regarding estrogen levels, some differences exist between SUM-159 and MDA-MB-453. In SUM159, E1 levels were decreased, while, in MDA-MB-453, E1 concentrations were increased. SUM-159 expresses at high intensity the ERβ in the nuclear and cytoplasmic membrane, denoting that SUM-159 may be more influenced by estrogen concentrations than MDA-MB-453. Thus, a decrease in E1 and DHT levels could favor cell viability reduction. On the other hand, although SUM-159 reduced cell viability with dutasteride, cell migration increased by 45% with respect to the control. Dutasteride treatment appeared to maintain E2 production in SUM-159 cells, suggesting that E2 concentrations promote cell migration. This is consistent with other authors that demonstrated that estrogens are involved in cellular migration processes [23,51].

The rest of the TNBC cell lines, SUM-149, MDA-MB-231 and MDA-MB-468, were not affected in terms of cell viability with dutasteride. SUM-149 and MDA-MB-231 correspond to those cell lines with the lowest expression of AR and Erβ; therefore, these cell lines may not be as influenced as MDA-MB-453 and SUM-159 in terms of cell viability. Regarding hormone secretion, in contrast to the reduction in DHT levels found in MDA-MB-453 and SUM159, DHT levels were practically not reduced in SUM-149, MDA-MB-231 and MDA-MB-468, denoting that cell proliferation may be influenced by the activation of AR and DHT concentrations. Indeed, dutasteride could partially inhibit 5αR in those cell lines, producing lower cell proliferation reduction. Nevertheless, SUM-149 and MDA-MB-468 reduced by approximately 15% cell migration with dutasteride. The estrogen secretion in both cell lines revealed a reduction in E1 levels, which may be related to the migration decrease found as both cell lines expressed ERβ in the cytoplasmic membrane; thus, estrogens could influence in migratory processes.

Therefore, the use of dutasteride reduces cell viability in TNBC cell lines with high expression of AR by reducing DHT secretion, which implies a lower AR activation and thus a decrease in cell viability. However, the use of anti-androgenic treatments could favor the estrogen secretion that may be involved in promoting cell migration processes.

It is well established that estrogens promote tumor development in TNBC [56]; thus, the use of aromatase inhibitors, such as anastrozole, may be a potential target for TNBC treatment. Similar to dutasteride results, SUM-159 and MDA-MB-453 were the two cell lines that reduced cell viability significantly with respect to the control. In hormonal terms, in both cell lines, a decrease in DHT levels was observed, corroborating that a decrease in androgen secretion affected cell proliferation. Although anastrozole inhibits estrogen production, it also produced an increase in A4 levels that may impair androgen metabolism, resulting in a decrease in DHT levels [57]. However, while in SUM-159 the E1 levels were decreased, in MDA-MB-453, they were increased. This could be due to differences in aromatase inhibition [30]. While in SUM-159 the aromatase inhibition was markedly in the formation of E1 from A4, in MDA-MB-453, the aromatase inhibition was mainly produced in the conversion of T to E2. Although few studies have been conducted on the genome encoding aromatase expression, it has been observed that, in breast cancer, the aromatase enzyme has single nucleotide polymorphisms (SNPs) that could interfere with the efficacy of anastrozole [58]. It is possible that some genetic variation in aromatase between cell lines produces the aromatase inhibition in A4 to E1 or T to E2, although more research is needed. In addition, Wang and colleagues (2010) have observed that the use of anastrozole produced higher E2 levels in breast cancer patients treated with aromatase inhibitors, suggesting that the existence of SNPs in aromatase may increase the aromatase activity despite the use of anastrozole [58].

In addition, it should be noted that SUM-149 and MDA-MB-468 were the only cell lines that obtained significant reductions in cell migration with anastrozole. In this case, anastrozole produces a decrease in estrogen levels in both cell lines. A reduction in E1 in SUM-149 and E2 in MDA-MB-468 could lead to a decrease in cell migration. In contrast, MCF-7 increases cell proliferation and migration with respect to the control, without altering estrogen levels. Numerous studies related the tumor progression increase under anti-aromatase treatments with the acquired resistance against anti-aromatase treatments in Erα- and PR-positive cell lines such as MCF-7 [55]. On the other hand, there is a significant increase in T levels in MCF-7, denoting that anastrozole treatment is more effective in the inhibition of the T to E2 conversion than in the inhibition of A4 to E1 conversion. Furthermore, the E1 increase found could be involved in the increase in cell viability by binding to ER [59].

Thus, the use of anastrozole reduces migration through estrogen depletion. However, either because of some acquired resistant mechanism to anti-aromatase treatments or because of SNPs that can be found in different breast cancer cell lines, anastrozole does not reduce estrogen levels in all TNBC cases. On the other hand, there was a cell proliferation decrease because the estrogen inhibition produces an increase in A4 levels and harms androgen metabolism, reducing DHT levels in TNBC cell lines. Thus, further studies on the presence of aromatase in TNBC are needed to improve treatment targeting aromatase inhibitors.

Finally, TNBC cell lines were less sensitive to the ASP9521 in terms of cell viability and migration. Results showed significant reductions in cell viability in SUM-159 and MDA-MB-453. ASP9521 inhibits the 17βHSD enzyme that converts A4 to T and E1 to E2 [60]. Both cell lines resulted in an increase in A4 levels and a reduction in E2 and T levels, denoting that treatment was efficient. The T and E2 deprivation produce less AR and ERβ activation and, therefore, a cell viability reduction. This fact is in line with other authors that showed the relation between AR and ERβ expression and tumor development [21,40]. In contrast, MCF-7 increase by approximately 10% the cell viability with respect to the control. In addition, an increase in E1 levels was observed, denoting that high estrogen concentrations promote cell viability and migration. Thus, the use of ASP9521 resulted in reduced T and E2 levels MDA-MB-453 and SUM-159, leading to decreased cell viability.

In summary, it has been observed that alterations in the steroid production and secretion pathways produced changes in cell viability and migratory processes in TNBC cells despite its receptor status. Although numerous studies have excluded hormonal therapies due to the lack of ERα, PR and HER-2 expression [35], the present study has demonstrated that TNBC could be also beneficiated by antihormone treatments due to expression of other hormone receptors such as AR and ERβ. Although this study has focused on the role of AR and ERβ, which are the two most controversial receptors in TNBC, it would be of interest to study whether the presence of GPER-1 could affect cell proliferation and migration processes. Estrogens, such as E2, could bind to GPER-1 and promote cell migration [37]. This study demonstrates that the expression intensity could determine the efficacy of the antihormonal treatments used since our results revealed that MDA-MB-453 AR- and Erβ-positive cells are more influenced by the use of anti-androgenic treatments than MDA-MB-231, which presented low AR and ERβ expression. This study also showed that the success of these treatments in AR- and Erβ-positive cells resides in a reduction in androgen and estrogen secretion that promotes a decrease either in cell viability or cell migration. On the other hand, MCF-7 reduced neither cell viability nor migration with the use of antihormonal treatments; thus, non-TNBC tumors could be more resistant to the use of these drugs than TNBC.

## 4. Materials and Methods

### 4.1. Cell Line Culture

Five human TNBC cell lines were studied: SUM-149, SUM-159, MDA-MB-231, MDA-MB-453 and MDA-MB-468. In addition, MCF-7, which express ERα and PR were used as a control [61]. SUM-149 and SUM-159 were originally obtained from Asterand, Plc. (Detroit, MI, USA) and were cultured in Nutrient mixture F-12 HAM medium (Sigma-Aldrich, St. Louis, MO, USA) supplemented with 5% charcoal-stripped fetal bovine serum (FBS) (Sigma-Aldrich, St. Louis, MO, USA), 1 µg/mL hydrocortisone (Sigma-Aldrich, St. Louis, MO, USA), 5 µg/mL insulin (Sigma-Aldrich, St. Louis, MO, USA) and 1% penicillin-streptomycin solution (Sigma-Aldrich, St. Louis, MO, USA). MDA-MB-231, MDA-MB-453 and MDA-MB-468 were obtained from ATCC (Virgina, USA) and cultured in RPMI1640 medium (Sigma-Aldrich, St. Louis, MO, USA) supplemented with 10% charcoal-stripped FBS (Sigma-Aldrich, St. Louis, MO, USA) and 1% penicillin–streptomycin solution (Sigma Aldrich, St. Louis, MO, USA). Finally, MCF-7, was obtained from ATCC (Manassas, Virginia, USA) and cells were cultured in DMEM high-glucose medium supplemented with 10% charcoal-stripped FBS (Sigma-Aldrich, St. Louis, MO, USA) and 1% penicillin–streptomycin solution (Sigma-Aldrich, St. Louis, MO, USA).

Cell lines were cultured in 25 cm^2^ culture flasks and maintained in a humidified 5% carbon dioxide atmosphere at 37 °C. The cell cultures were observed daily by phase-contrast microscope (Optika XDS-2 Inverted Microscope, Euromicroscopes, S.L, Barcelona, Spain) in order to check cell viability and growth.

### 4.2. Treatments

Dutasteride and anastrozole were obtained from Sigma-Aldrich (St. Louis, MO, USA). Finally, ASP9521 was obtained from Adooq bioscience (Irvine, CA, USA). All drugs were diluted in dimetil-sulfoxide (DMSO) (Sigma Aldrich, St. Louis, MO, USA) to start from an initial concentration of 20 mM, 34 mM and 30 mM for dutasteride, anastrozole and ASP9521, respectively. Then, drugs were diluted again to achieve a work concentration of 10 mM and 1 mM for all treatments.

### 4.3. Immunofluorescence

A total of 5 × 10^4^ cells were plated on culture chambers (Sarstedt, Germany), fixed with 4% paraformaldehyde (Invitrogen, Waltham, MA, USA), and permeabilized with 0.5% Triton X-100 (Sigma-Aldrich, St. Louis, MO, USA). Then, cells were blocked with goat serum and subsequently incubated with rabbit primary AR monoclonal antibody (MA5-13426; Sigma-Aldrich, St. Louis, MO, USA) and rabbit primary ERβ polyclonal antibody (PA1-310B; Invitrogen, Massachusetts, USA) overnight at 4 °C and with shaking. Then, cells were washed with PBS and incubated with secondary antibody (CF488A; Sigma-Aldrich, St. Louis, MO, USA). Finally, slides were mounted with Prolong Gold Antifade with DAPI (Invitrogen, Waltham, MA, USA). Images were captured with the Optika fluorescence microscope (Optika IM-3LD2 Microscope, Bérgamo, Italy). Immunofluorescence quantification was analyzed with the RGB value plugin available in ImageJ software 1.53e version.

### 4.4. Sensitivity Assay

The sensitivity assay was carried out in order to determine the half maximal effective concentration (EC_50_) of dutasteride, anastrozole and ASP9521 in vitro [24]. The assay was realized in duplicate. Each cell line was cultured in 96-well polystyrene plates (Corning Incorporated, New York, NY, USA) at a density of 1 × 10^3^ cells per well. Then, cell lines were treated with 5-fold serial dilutions (from 10 mM to 64 nM) of each treatment. All drugs were diluted in DMSO and a control (cell lines only with DMSO) was performed to consider the possible toxicity generated by the compounds.

Cells were incubated for 72 h in a humidified 5% carbon dioxide atmosphere at 37 °C. Then, bromide of 3-(4, 5-dimetiltiazol-2-ilo)-2,5-difeniltetrazol (MTT) (Sigma-Aldrich, St. Louis, MO, USA) was added in all wells. Finally, the absorbance was measured at a wavelength of 568 nm with an automatic plate reader (ThermoFisher Scientific, Waltham, MA, USA). The results were processed with GraphPad Prism 6.01 software to obtain the EC-50 of each drug.

### 4.5. Cell Viability Assay

Each cell line was seeded in 96-well polystyrene plates (Corning Incorporated, New York, NY, USA) at a density of 1 × 10^3^ cells per well in each supplemented culture medium with 1 μM dutasteride, anastrozole and ASP9521. The assay was carried out in duplicate and untreated cells were used as a control. Cells were cultured in a humidified 5% CO_2_ atmosphere at 37 °C for 24 h. Then, the viability assay was performed by adding MTT and measuring the absorbance at 568 nm. The data were expressed as a percentage of cell viability with respect to the control cells.

### 4.6. Wound Healing Assay

A total of 1 × 10^5^ cells per well were plated in 24-well polystyrene plates (Corning Incorporated, New York, USA). When cells reached a confluence of 90%, a wound was performed in the middle of the well and the treatments were added at a concentration of 1 µM. The cells were cultured in a humidified 5% CO_2_ atmosphere at 37 °C for 24 h. Images were taken for each well at the time the wound was performed and after 24 h with a phase-contrast microscope (Optika XDS-2 Inverted Microscope, Euromicroscopes, S.L, Spain). In addition, culture media were collected at 24 h for hormonal analyses.

Images were processed by ImageJ MRI-Wound Healing Tool software 1.53e version, comparing the wound width at zero and at 24 h of the different treatments with respect to the control. Measures were obtained in pixels and represented as a percentage of wound closure with respect to the control.

### 4.7. Steroid Determinations in Culture Media

Estrone (E1), 17β-estradiol (E2), testosterone (T) and androstenedione (A4) antibodies were developed in the Department of Animal Physiology (UCM, Madrid, Spain). To determine these hormones in culture media samples, an amplified EIA previously validated was performed [62]. Dihydrotestosterone (DHT) and dihydroepiandrostenedione (DHEA) determinations were performed using a commercially available EIA kit (DE5761 and DEH3344, respectively) (Demeditec, Kiel, Germany) according to manufacturer’s instructions. The hormones determined and the antibodies used are summarized in Table 1.

For amplified EIA, 96-well flat-bottom medium-binding polystyrene microplates (Biohit, Helsinki, Finland) were coated overnight at 4 °C with the appropriate purified antibody dilutions. The next day, plates were washed and standards and culture samples were added in duplicate and incubated overnight at 4 °C. On the last day, conjugate working solutions (CWS) were added to each well and plates were incubated for 4 h at room temperature. After washing, Enhance K-Blue TMB substrate (Neogen, Lexington, KY, USA) was added to evaluate the amount of labelled steroid hormones. Finally, the colorimetric reaction was stopped by the addition of 10% H_2_SO_4_ to each well. Absorbance was read at 450 nm using a 96-well SpectraMax 190UV/Vis automatic plate reader (Eurogenetics, Seraing, Belgium). Hormone concentrations were calculated by means using a software developed for this technique (ELISA AID, Eurogenetics, Seraing, Belgium). A standard dose–response curve was constructed by plotting the binding percent (B/B0 × 100) against each steroid hormone standard concentrations. All hormone concentrations were expressed in ng/mL, except for DHT culture media hormone concentrations, which were expressed in pg/mL.

### 4.8. Statistics

Data were analyzed by IBM SPSS Statistics 25 software. In all statistical comparisons, *p* values < 0.05 were considered statistically significant. One-way analysis of variance (ANOVA), specifically Bonferroni test, was used to compare experimental groups with respect to the control in cell viability, wound healing assays and culture media enzyme immunoassays. In addition, U Mann–Whitney test was used to compare the mean immunofluorescence intensity.

## 5. Conclusions

Numerous authors have excluded the use of antihormonal therapies for TNBC treatment. However, this study confirms that steroid hormones exert an important role in the processes of cell viability and migration in vitro and antihormonal treatments could be beneficial for TNBC treatment by activating AR and ERβ pathways. The androgen and estrogen depletion produces a cell viability and migration reduction on TNBC AR- and Erβ-positive cell lines. This study is useful to better understand the role of steroid hormones in TNBC and the importance of studying the expression of different hormone receptors, such as AR and ERβ, in addition to the conventional ones currently used for breast cancer classification.

## Figures and Tables

**Figure 1 ijms-25-01471-f001:**
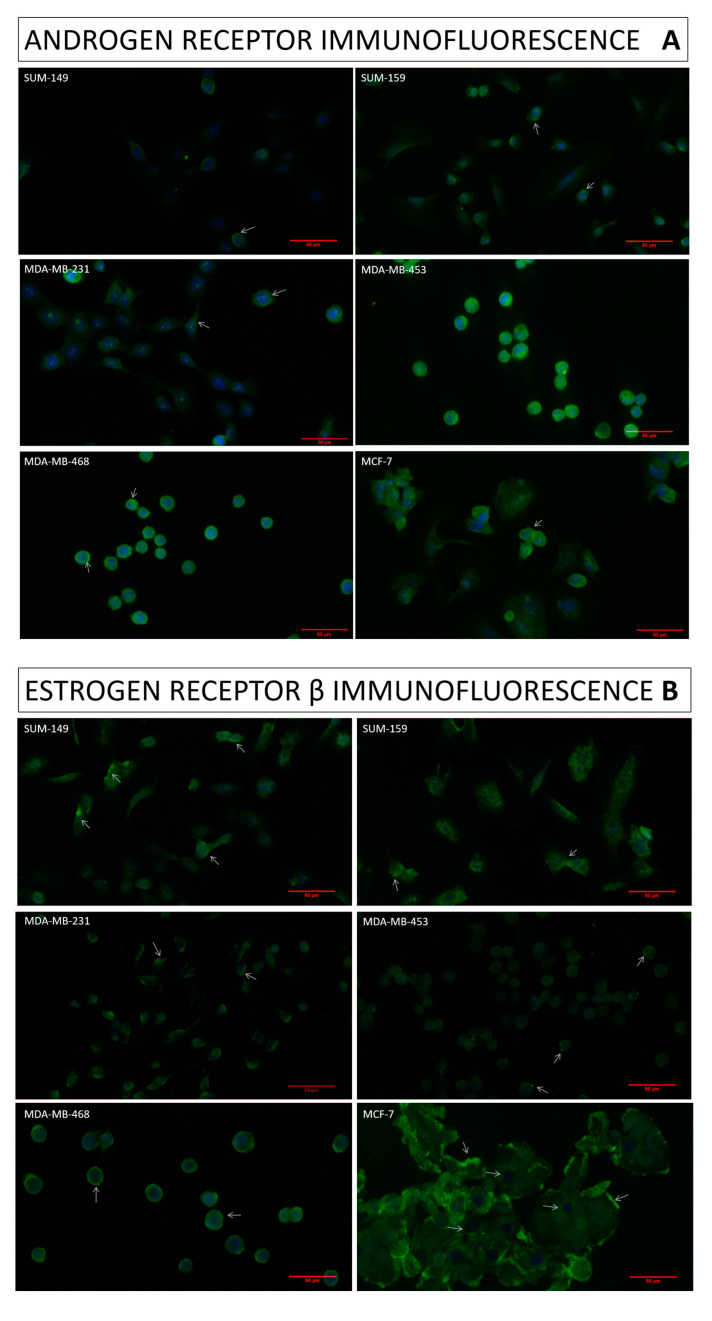
Immunofluorescence results confirm the expression of the androgen receptor (AR) (**A**) and estrogen receptor β (ERβ) (**B**) in all cell lines (green). Cells were DAPI-stained to visualize cell nuclei (blue). White arrows indicate the localization of the highest expression of AR and ERβ in each cell line; in addition, they indicate the low nuclear expression of ERβ in MCF-7. Quantification of AR and ERβ immunofluorescence in all studied cell lines (mean of green value ± SD); significant differences (*p* < 0.001) between cell line are indicated by an asterisk (*); differences were found between *1 (high expression of AR or ERβ) and *2 (low expression of AR or ERβ) for each receptor (**C**). Images were taken at 40× magnification. The merge and the quantification were carried out using ImageJ software 1.53e version.

**Figure 2 ijms-25-01471-f002:**
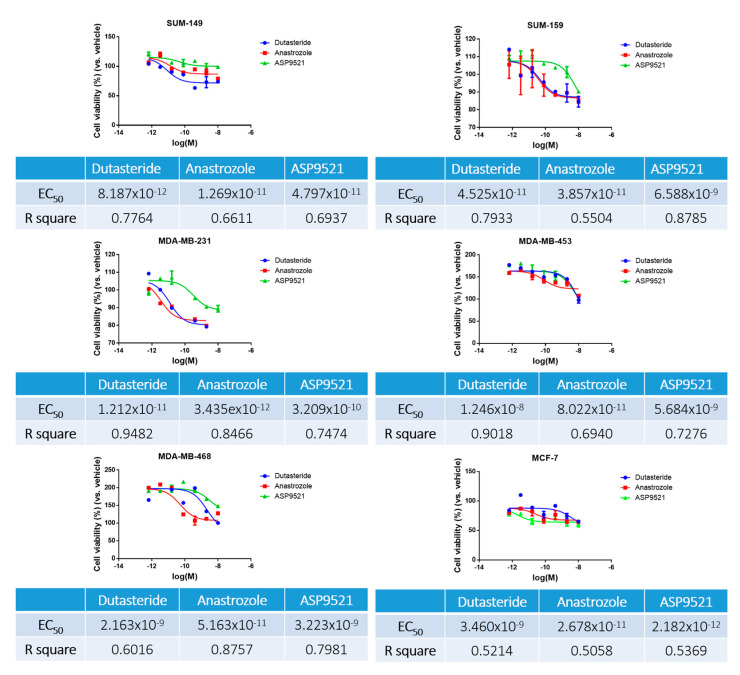
Sensitivity assay results of dutasteride, anastrozole and ASP9521 in TNBC cell lines. Results are expressed graphically with cell viability percentage ± standard deviation (SD) as a function of the molarity logarithm (Log (M)) of each drug. In addition, results are expressed in a table where the EC_50_ and their R-square are indicated.

**Figure 3 ijms-25-01471-f003:**
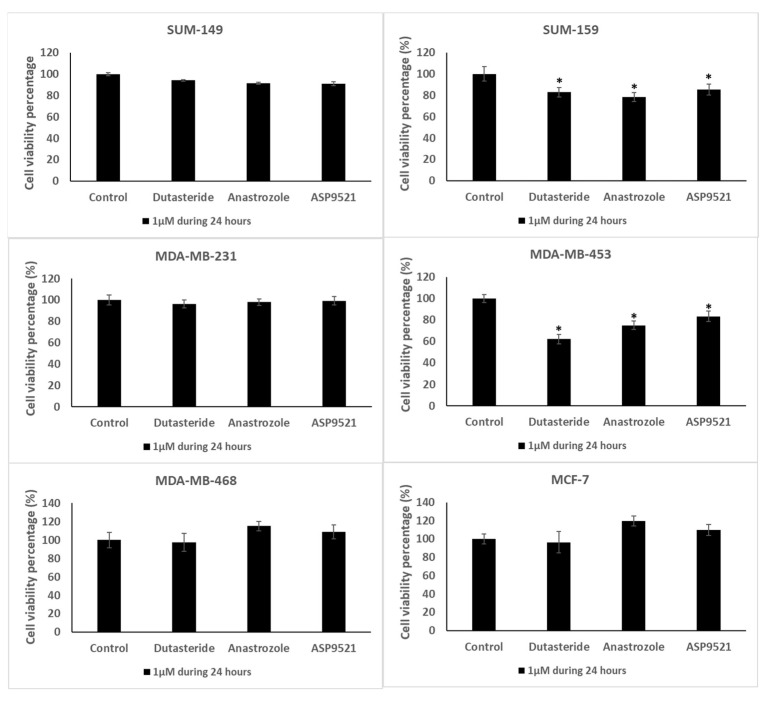
Cell viability results of dutasteride, anastrozole and ASP9521 in TNBC and MCF-7 cultured cell lines. Values are represented as cell viability percentage ± SD. * Denoted significant differences (*p* < 0.05) between control and treatments.

**Figure 4 ijms-25-01471-f004:**
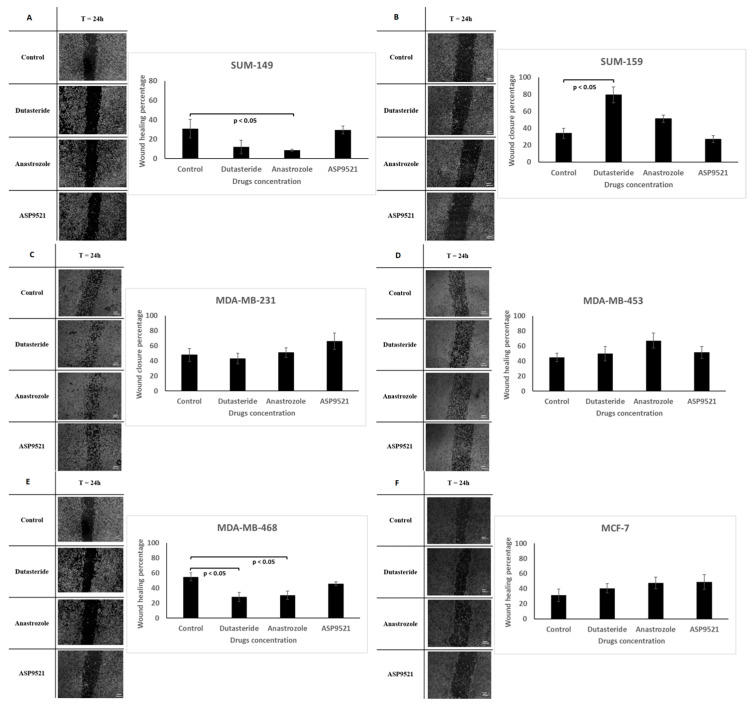
Cell migration (wound healing assay). Images were recorded 24 h after performing the wound. Representative images are shown from dutasteride, anastrozole and ASP9521 treatments and the control at different cell lines: SUM-149 (**A**), SUM-159 (**B**), MDA-MB-231 (**C**), MDA-MB-453 (**D**), MDA-MB-468 (**E**) and MCF-7 (**F**). The wound area is the black area without cells. Images were processed with ImageJ software 1.53e version with a scale bar of 100 µm. Values are represented as percentage wound closure ± SD. Significant differences (*p* < 0.05) between control and treatments are marked.

**Figure 5 ijms-25-01471-f005:**
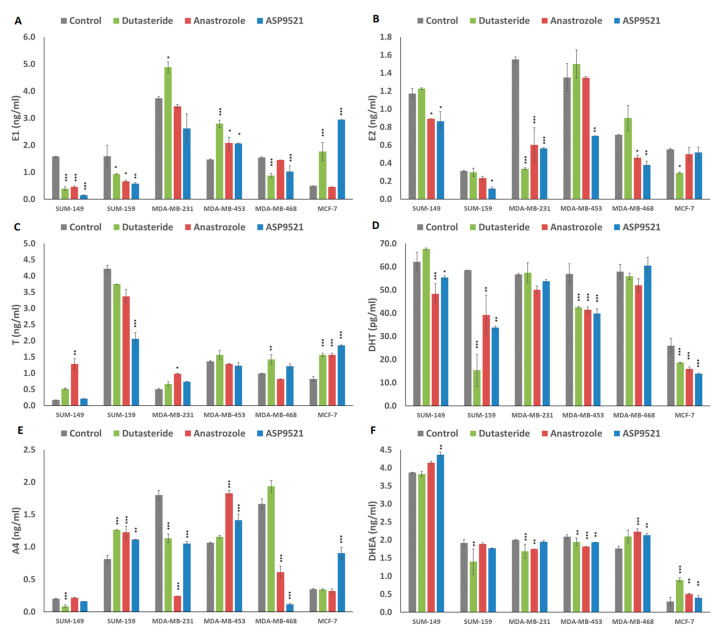
Graphs represent hormone secretion concentration in cultured media of cells treated with dutasteride, anastrozole and ASP9521 in each cell line. Bars represent means ± SD. (**A**) Estrone (E1) (ng/mL), (**B**) 17β-estradiol (E2) (ng/mL), (**C**) testosterone (T) (ng/mL), (**D**) dihydrotestosterone (DHT) (pg/mL), (**E**) androstenedione (A4) (ng/mL) and (**F**) dihydroepiandrostenedione (DHEA) (ng/mL) levels were determined. *p* < 0.05 (*), *p* = 0.01 (**), *p* < 0.001 (***) denote significant differences between control and treatments.

**Table 1 ijms-25-01471-t001:** Steroid hormones assayed and antibodies used for EIA determinations. DHT and DHEA were determined using commercial kits following manufacturer’s instructions.

Hormone	Abbreviation	Antibody Code	Dilution
Estrone	E1	R522-2	1/12,000
17β-estradiol	E2	C6-E91	1/4000
Testosterone	T	R156	1/8000
Androstenedione	A4	C9111	1/5000
Dihydrotestosterone	DHT	DE5761	
Dihydroepiandrostenedione	DHEA	DEH3344	

## Data Availability

Data are contained within the article.

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
