# Peer review of "Androgen and Estrogen β Receptor Expression Enhances Efficacy of Antihormonal Treatments in Triple-Negative Breast Cancer Cell Lines"

_ijms, 2024, doi:10.3390/ijms25031471_

Round 1
Reviewer 1 Report
Comments and Suggestions for Authors
In this manuscript the authors investigate whether TNBC cell lines might be sensitive to anti-hormonal treatment. This hypothesis is interesting and not well studied up to now. Indeed TNBC cells express “alternative” receptors for estrogen (ESR2, GPER1) and androgens as well. So this is an interesting paper for the readers of this journal.
The authors analyzed 5 TNBC cell lines and MCF-7 as control. What was the rational for choosing these lines? Do these cell lines represent distinct TNBC subtypes?
Before publication, however, please clarify the following points:
Please also discuss the influence GPER1 might have on anti-hormonal treatments in these cell lines.
Usually phenol red is avoided when doing estrogen research, as it or its derivatives can mimic an estrogen effect. Please comment!
Immuno fluorescence looks convincing but as these data are rather qualitative please show Westerns and may be also mRNA data as well.
In fig. 2, I do not really understand why cell viability (in % of vehicle) is often significantly higher than 100%. Please explain and discuss. Cell viability in contrast to sensitivity is only defined by a different time point of analysis (24h v. 72h) and concentration, which was apparently fixed for viability assays to 1 µM. The rational for this should be better explained. Altogether it is unclear what the cause for the sensitivity is. Do the cells die, e.g. by apoptosis, is it cell cycle arrest, is it loss of viability? Please perform appropriate assays to answer this question.
Fig. 3 the X-axis is named “drug concentration”. But no concentration is given. Please add concentration and time to the figure legend for clarification. The same problem occurs in Fig. 4 (scratch assay).
What about proliferation during the scratch assay? Can you exclude that this contributes to scratch closure?
The determination of hormone levels in the cell culture is a very useful experiment. Providing antibody concentrations used in the assays would be helpful. Here I am also wondering to what extend the FBS contributes to the determined concentrations. Performing this with char coal stripped FBS might strengthen your results.
Fig. 5: Please remove the “,” and use “.” instead. You might also fuse the left and right part into one single graph.
Comments on the Quality of English Language
The paper is generally well written and needs only a bit of language polishing.
Author Response
REVIEWER 1
In this manuscript the authors investigate whether TNBC cell lines might be sensitive to anti-hormonal treatment. This hypothesis is interesting and not well studied up to now. Indeed TNBC cells express “alternative” receptors for estrogen (ESR2, GPER1) and androgens as well. So this is an interesting paper for the readers of this journal.
Thank you very much for your appreciation. We are glad to read that you found it interesting research.
The authors analyzed 5 TNBC cell lines and MCF-7 as control. What was the rational for choosing these lines? Do these cell lines represent distinct TNBC subtypes?
The TNBC cell lines used are those available at the research center. The use of these five cell lines has been considered because they belong to different TNBC subtypes.
- SUM-149 is a ductal carcinoma of inflammatory basal-like 2 triple negative (BSL-2 – TNBC) cell line.
- SUM-159 is a non-inflammatory mesenchymal stem like triple negative (MSL – TNBC) carcinoma cell line.
- MDA-MB-231 is a mesenchymal stem like triple negative (MSL – TNBC) adenocarcinoma cell line.
- MDA-MB-453 is a triple negative luminal androgen receptor (TNBC-LAR) cell line. Considered by numerous investigators for the study of the role of androgen receptor in triple negative breast cancer.
- MDA-MB-468 is an adenocarcinoma of basal like triple negative (BSL-1 – TNBC) cell line.
In addition to the different TNBC subtypes, these cell lines are commonly used in TNBC cancer research.
Before publication, however, please clarify the following points:
Please also discuss the influence GPER1 might have on anti-hormonal treatments in these cell lines.
We have added the presence of GPER-1 in TNBC at line 257 as follows: “These hormones may exert their action by binding to other receptors such as the AR, or ERβ or GPER-1, among others [8, 36]. GPER-1 is present in half of TNBCs and its presence has been associated with a metastasis increase and worse prognosis [37, 38].”
Also, we have discussed the influence of GPER-1 at line 417 as follows: “Although this study has focused on the role of AR and ERβ, which are the two most controversial receptors in TNBC, it would be of interest to study whether the presence of GPER-1 could affect cell proliferation and migration processes. Estrogens, such as E2, could bind to GPER-1 and promote cell migration [37].
Usually phenol red is avoided when doing estrogen research, as it or its derivatives can mimic an estrogen effect. Please comment!
Thank you very much for your appreciation. The presence of phenol red in the culture medium was one of the factors that concerned us because of the possibility of its interference with estrogens. After performing literature research, it has been found that phenol red has no interference on TNBC cell lines (Katzenellenbogen et al, Proc Natl Acad Sci U S A. 1986. 83(8):2496-2500). Besides, we have observed that the articles that discuss the interference of phenol red in cell cultures do at a concentration of 20-30μM (Liu et al, PLoS One. 2013. 8(4): e60189), in our cultures the concentration of phenol red is around 0.015μM. Therefore, the concentration of phenol red in our culture medium is much lower than the concentration that can affect cell lines.
Immuno fluorescence looks convincing but as these data are rather qualitative please show Westerns and may be also mRNA data as well.
Thank you very much for the suggestion, we believe that immunofluorescence results are very illuminating. However, they are so qualitative. The other reviewer concurs with your assessment and recommends quantifying the intensity of immunofluorescence. Therefore, we have decided to quantify immunofluorescence using the RGB plugin accessible in ImageJ software. In addition, we have carried out statistics of the results obtained. With all this, our results are largely supported and increase the quality of the assay. Western blot tests have not been performed since we believe that the cost of the technique is not necessary, having the possibility of performing quantification with ImageJ.
We have added the graph to the immunofluorescence results section (section 2.1). In addition, we have added the procedure to the material and methods (section 4.3) and we have indicated the statistical test used in section 4.8.
In fig. 2, I do not really understand why cell viability (in % of vehicle) is often significantly higher than 100%. Please explain and discuss. Cell viability in contrast to sensitivity is only defined by a different time point of analysis (24h v. 72h) and concentration, which was apparently fixed for viability assays to 1 µM. The rational for this should be better explained. Altogether it is unclear what the cause for the sensitivity is. Do the cells die, e.g. by apoptosis, is it cell cycle arrest, is it loss of viability? Please perform appropriate assays to answer this question.
Thank you for your interest. The sensitivity assay aims to determine the half-maximal effective concentration (EC-50) of a drug. To obtain this value, it is necessary to dilute the drug at different concentrations. Typically, drugs are dissolved in DMSO, which can be toxic to cells. To prevent this toxicity, the concentration of DMSO in the extracellular medium is always less than 0.01%. To rule out the toxicity effect of DMSO on cells and to use it as a control (vehicle), the entire top row of wells (row A of the polystyrene plates) were treated only with DMSO. The control is taken as 100% cell viability. Thus, the cell viability results obtained with the different concentrations are put in percentage with respect to the vehicle. At extremely low doses of the drug, it appears that the cells are not affected and therefore tend to proliferate more even than with DMSO alone. For this reason, cell viability (vs vehicle) starts a higher concentration than 100%. Probably, cells at low drug concentrations can promote proliferation to ensure cell survival, but at an effective drug dose (EC-50), cell proliferation decrease. In the sensitivity assay, MTT is a compound that binds to the nuclei of living cells, therefore, what is measured in the sensitivity assay is cell viability.
On the other hand, the cell viability assay is performed after having obtained the EC-50 in the cell sensitivity assay. The purpose of the sensitivity assay is to obtain a single effective dose for all cell lines and the different treatments (dutasteride, anastrozole and ASP9521). The cell viability assay aims to study cell proliferation under the chosen effective dose and is carried out with an incubation period of 24 hours, the same time as the migration assay in order to see how the drugs affect in terms of proliferation and migration under the same time of exposure and dose of the drug.
Finally, the reason why TNBC cells are sensitive to the anti-hormonal treatments is one of the pillars described in the article. We supposed that the hormonal variations under treatment are the main cause of the cell viability decrease and, as mentioned, cell sensitivity is a cell viability with a different objective and greater number of doses tested to find the effective dose.
Fig. 3 the X-axis is named “drug concentration”. But no concentration is given. Please add concentration and time to the figure legend for clarification. The same problem occurs in Fig. 4 (scratch assay).
Thank you for the recommendation. We have made the changes suggested in figure 3 and 4.
What about proliferation during the scratch assay? Can you exclude that this contributes to scratch closure?
Researchers commonly use the wound closure assay to study cell migration processes in vitro due to its simplicity and cost-effectiveness. To minimize the influence of proliferation in this assay, it is recommended to use the minimum amount of culture medium necessary for cell survival. Additionally, it is advised to limit the incubation time to 24 hours or less to prevent excessive cell proliferation.
Besides, there are morphological differences between cells at the wound boundary and those at the sides of the well. Specifically, the cells at the wound boundary exhibit a more elongated conformation, which is consistent with a prolongation of the cytoskeleton to close the wound.
The determination of hormone levels in the cell culture is a very useful experiment. Providing antibody concentrations used in the assays would be helpful. Here I am also wondering to what extend the FBS contributes to the determined concentrations. Performing this with char coal stripped FBS might strengthen your results.
Thank you very much for your comment. The antibody concentrations used are described in Table 1 on page 16. Regarding the concentration of steroid hormones in the culture medium, our research center has been developing this technique for several years (Illera et al, Horm Mol Biol Clin Invest 2015; 24(3): 137–145). The hormone measurements were performed in medium supplemented with charcoal-stripped, we have already corrected the mistake in section 4.1.
Fig. 5: Please remove the “,” and use “.” instead. You might also fuse the left and right part into one single graph.
Thank you for the recommendation. We have made the changes suggested in figure 5.

Reviewer 2 Report
Comments and Suggestions for Authors
The triple negative breast cancer (TNBC) subtype is defined by the absence of estrogen receptor α (ERα), progesterone receptor (PR), and HER-2 overexpression; however, TNBC can express the androgen receptor (AR) or estrogen receptor β (ERβ) and secrete steroid hormones. Therefore, inhibition of steroid hormone secretion may represent a viable therapeutic approach for TNBC treatment. This study aimed to assess the efficacy of dutasteride, anastrozole, and ASP9521 in in vitro models using human TNBC cell lines. Immunofluorescence, sensitivity, proliferation, and wound healing assays were conducted, and hormone concentrations were measured. The results indicated that all TNBC cell lines expressed AR and ERβ, with higher intensity expression correlating with greater sensitivity to anti-hormonal treatments. All treatments reduced cell viability, particularly in MDA-MB-453 and SUM-159 cells. Notably, a decrease in androgen levels was observed in these cell lines, which may contribute to the reduction in cell viability. Conversely, MCF-7 and SUM-159 cells exhibited increased cell migration under treatment, accompanied by elevated estrogen levels, potentially promoting cell migration. Overall, these findings suggest that anti-hormonal treatments may hold promise for TNBC therapy, and highlight the importance of steroid hormones in AR and ERβ positive TNBC cell lines. This is an interesting and informative manuscript, but I have several following concerns:
1.The authors would better calculate the mean fluorescence intensity of the expression of the two receptors in each cell line and perform statistical analysis to better compare the differences in the expression of the two proteins in different cell lines.
2. The authors studied only one non-triple-negative breast cancer cell line and make the conclusion that androgen and estrogen β receptors expression enhance efficacy of anti-hormonal treatments in triple negative breast cancer cell lines. More evidence is needed to support this conclusions. It is recommended to add several non-triple-negative breast cancer cell lines, such as T47D, Hs578...
3. Authors are advised to adjust Figures 4 and 5 below to allow them to lay out the layout of the paper.
4. A "Discussion" is imperative to clarify how the authors' new findings differ from previous studies and potential applications.
5. Abbreviations should be defined when they first appear in the text. Such as "DHEA" in Line 73,...
6. There are many errors in the superscript and subscript. Please double check and correct them. For example "1x10 -5 M to 6.4x10 -10 M" in line 131 and EC50 in line 133.
7. Some punctuation marks should have Spaces, check carefully and add.
8. "p" denoting statistical significance should be italicized.
9. The nucleic acid sequences (including gene names, regulatory sequences, and primer names) should be in italics. Such as "TP53" in line 253...
10. Please unify the format of references in the article, including the author's name, the case of words in the title of the article, the writing of the name of the journal, and the page number.
Comments on the Quality of English LanguageMinor editing of English language required.
Author Response
The triple negative breast cancer (TNBC) subtype is defined by the absence of estrogen receptor α (ERα), progesterone receptor (PR), and HER-2 overexpression; however, TNBC can express the androgen receptor (AR) or estrogen receptor β (ERβ) and secrete steroid hormones. Therefore, inhibition of steroid hormone secretion may represent a viable therapeutic approach for TNBC treatment. This study aimed to assess the efficacy of dutasteride, anastrozole, and ASP9521 in in vitro models using human TNBC cell lines. Immunofluorescence, sensitivity, proliferation, and wound healing assays were conducted, and hormone concentrations were measured. The results indicated that all TNBC cell lines expressed AR and ERβ, with higher intensity expression correlating with greater sensitivity to anti-hormonal treatments. All treatments reduced cell viability, particularly in MDA-MB-453 and SUM-159 cells. Notably, a decrease in androgen levels was observed in these cell lines, which may contribute to the reduction in cell viability. Conversely, MCF-7 and SUM-159 cells exhibited increased cell migration under treatment, accompanied by elevated estrogen levels, potentially promoting cell migration. Overall, these findings suggest that anti-hormonal treatments may hold promise for TNBC therapy, and highlight the importance of steroid hormones in AR and ERβ positive TNBC cell lines. This is an interesting and informative manuscript, but I have several following concerns:
1.The authors would better calculate the mean fluorescence intensity of the expression of the two receptors in each cell line and perform statistical analysis to better compare the differences in the expression of the two proteins in different cell lines.
Thank you very much for the recommendation, we believe that this suggestion reinforces our results. We have added the graph to the immunofluorescence results section (section 2.1). In addition, we have added the procedure to the material and methods (section 4.3) and we have indicated the statistical test used in section 4.8.
- The authors studied only one non-triple-negative breast cancer cell line and make the conclusion that androgen and estrogen β receptors expression enhance efficacy of anti-hormonal treatments in triple negative breast cancer cell lines. More evidence is needed to support this conclusions. It is recommended to add several non-triple-negative breast cancer cell lines, such as T47D, Hs578...
The aim of this article is to study the effect of anti-hormonal treatments on TNBC cell lines. MCF-7 has been used as a control to observe the comparison between TNBC and a hormone receptor positive cell line in terms of hormonal secretion. Therefore, we believe that since this is not the main objective of the article, more than one non-TNBC cell line is not necessary. Nevertheless, we will consider it for further studies.
- Authors are advised to adjust Figures 4 and 5 below to allow them to lay out the layout of the paper.
Thank you for the annotation. We have modified figures 4 and 5 according to the characteristics required by the journal. We hope there will be no problems in the layout.
- A "Discussion" is imperative to clarify how the authors' new findings differ from previous studies and potential applications.
Throughout the discussion, an extensive bibliography was used to support the obtained results. The last paragraph of the discussion and the conclusions at the end of the article present the new findings of this study, confirming that the studied anti-hormonal treatments can be useful for decreasing proliferation and migration in vitro, despite the fact that numerous authors have not considered them for TNBC.
This study shows that high expression of AR and ERβ is associated with a better response to anti-hormonal therapies, as observed in MDA-MB-453 and SUM-159 cell lines. Conversely, cell lines with low expression of AR and ERβ, such as MDA-MB-231, exhibit a lower response to these therapies. Furthermore, this study confirms that changes in cell proliferation and migration are caused by alterations in the steroid pathway. Therefore, the presence of not only the “main” receptors (ERα, PR and HER-2) influences in tumor progression, but also the hormonal secretion of TNBC cell lines could promote cell proliferation and migration.
We believe that the discussion is supported by other researchers to reinforce the results obtained, concluding with a final paragraph summarizing the contribution of this article to the scientific community. In addition, the usefulness and novelty of this study is highlighted in the conclusions. However, if there is any specific aspect that you believe should be added to this section, it will be taken into account.
- Abbreviations should be defined when they first appear in the text. Such as "DHEA" in Line 73,...
Thank you for your comment. DHEA was defined the first time it appeared in the text. However, we have reviewed the entire article and added “3β-hydroxysteroid dehydrogenase (3βHSD)” in line 74 and we have changed “17β-hidroxiesteroide deshidrogenase 5” in line 96 to “17β-hydroxysteroid dehydrogenase (17βHSD)” in line 75 because it is the first time that appears in the text.
- There are many errors in the superscript and subscript. Please double check and correct them. For example "1x10 -5 M to 6.4x10 -10 M" in line 131 and EC50 in line 133.
Thank you very much for the warning. We have reviewed and corrected all mistakes made with superscripts and subscripts. As well as modified in Figure 2.
- Some punctuation marks should have Spaces, check carefully and add.
We have changed “,” to “.” In line 116, added a space after “.” in line 119 and removed a space after “.” in line 181.
Also, we have changed “anti-aromatase drugs are recommended by the American Society of Clinical Oncology (ASCO) as initial adjuvant therapy” to “American Society of Clinical Oncology (ASCO) as initial adjuvant therapy recommends the anti-aromatase drugs” in order to improve English language.
- "p" denoting statistical significance should be italicized.
Thank you very much for the clarification. We have corrected all p values ​​and put them in italics.
- The nucleic acid sequences (including gene names, regulatory sequences, and primer names) should be in italics. Such as "TP53" in line 253...
Thank you very much for the clarification, we have corrected the mistake.
- Please unify the format of references in the article, including the author's name, the case of words in the title of the article, the writing of the name of the journal, and the page number.
We have reviewed the bibliography and corrected the mistakes we have found, thank you for your comment.

Round 2
Reviewer 2 Report
Comments and Suggestions for Authors
The authors have addressed all my concerns. I recommend accepting it in current form.